# Going “Green” in the Prevention and Management of Atherothrombotic Diseases: The Role of Dietary Polyphenols

**DOI:** 10.3390/jcm10071490

**Published:** 2021-04-03

**Authors:** Ana Reis, Sara Rocha, Victor de Freitas

**Affiliations:** REQUIMTE-LAQV, Department of Chemistry and Biochemistry, Faculty of Sciences, University of Porto, Rua do Campo Alegre, 4169-007 Porto, Portugal; up201502921@fc.up.pt (S.R.); vfreitas@fc.up.pt (V.d.F.)

**Keywords:** mediterranean diets, CVD, lipoproteins, appetite, obesity, hypertension, inflammation, gut metabolites, vascular health

## Abstract

During the 20th century processed and ready-to-eat foods became routinely consumed resulting in a sharp rise of fat, salt, and sugar intake in people’s diets. Currently, the global incidence of obesity, raised blood lipids, hypertension, and diabetes in an increasingly aged population contributes to the rise of atherothrombotic events and cardiovascular diseases (CVD) mortality. Drug-based therapies are valuable strategies to tackle and help manage the socio-economic impact of atherothrombotic disorders though not without adverse side effects. The inclusion of fresh fruits and vegetables rich in flavonoids to human diets, as recommended by WHO offers a valuable nutritional strategy, alternative to drug-based therapies, to be explored in the prevention and management of atherothrombotic diseases at early stages. Though polyphenols are mostly associated to color and taste in foods, food flavonoids are emerging as modulators of cholesterol biosynthesis, appetite and food intake, blood pressure, platelet function, clot formation, and anti-inflammatory signaling, supporting the health-promoting effects of polyphenol-rich diets in mitigating the impact of risk factors in atherothrombotic disorders and CVD events. Here we overview the current knowledge on the effect of polyphenols particularly of flavonoid intake on the atherothrombotic risk factors and discuss the caveats and challenges involved with current experimental cell-based designs.

## 1. Introductory Perspective and Focus

The 20th century brought massive changes to people’s lifestyle and eating habits. Processed and ready-to-eat foods became routinely consumed resulting in a sharp rise of fat, salt, and sugar intake in people’s diets. In consequence, the global incidence of obesity, blood lipids, hypertension, and diabetes in the world’s population has escalated in the past decades leading to an increased prevalence of cardio- and vascular- diseases and related morbidity with atherothrombotic diseases. According to the World Health Organization (WHO), cardio- and vascular diseases are already responsible for 17.5 million (31%) deaths worldwide [1]. In recent years, deaths by non-communicable diseases have decreased in Western countries due to improvement of healthcare services; however, in the past four decades the worldwide obesity has tripled, the prevalence of diabetes among adults (>18 yr) has nearly doubled and one in seven adults have hypertension [2]. In EU alone, health expenditure with diet-related diseases is steadily increasing and accounted for 7.8% of GDP in 2017 [3]. The deleterious impact of atherothrombotic diseases is likely to rise in the future with (i) the expected ageing of the population; (ii) the co-morbidity among the older population, where around 50 million EU citizens are over 65 years old and estimated to suffer from two or more chronic conditions [3]; and (iii) most significantly with the noticeable incidence of obesity and diabetes among children and young adults [2].

The dramatic socio- and economic impact of atherothrombotic disorders on our society is largely due to its asymptomatic nature with the “silent” onset at early age and progress to plaque deposition in adulthood with detrimental impact to cerebral, coronary and peripheral vasculature. At the moment, drug-based therapies are a valuable strategy to tackle and help manage elevated blood lipids, raised blood pressure, and high blood sugar. Individually most synthetic drugs are well-tolerated by the general population resulting in an overall positive benefit/risk balance, but all drugs have adverse side effects particularly if used in combination with others. For instance, statins are routinely prescribed but the long-term effects remain under strict scrutiny [4,5,6]. Moreover, with the increasing incidence of these pathologies in the general population and particularly in children and young adults there are growing ethical concerns on whether these drugs should be broadly prescribed to children considering the known side effects and the still unknown long-term effects.

The adoption of healthier lifestyle and dietary choices with the inclusion of fresh and unprocessed fruits, vegetables, nuts, and seeds rich in polyphenols typical of Mediterranean and Nordic style Diets, as recommended by the World Health Organization (WHO), could mitigate the deleterious effects of diet-related diseases and contribute to the sustainability of national Health systems.

Polyphenols found in fruits, vegetables, nuts, seeds, and spices are a diverse group of natural compounds with more than 8000 structures already identified ranging from low molecular weight phenolic acids to high molecular weight proanthocyanidins [7,8,9]. Among the thousands of food polyphenols, these are typically divided in two main categories (Appendix A), namely, (1) flavonoid-based compounds that include flavanols, flavones, flavanones, and anthocyanins; and (2) non-flavonoid compounds that include stilbenes, hydroxybenzoic acids, hydroxycinnamic acids, and tannin derivatives [7,8,9]. Polyphenols in foods are responsible for organoleptic properties such as color and taste [10,11] and particularly for conferring the bitter taste and astringent sensation in fresh unripe foods [11,12,13]. The astringency associated to polyphenol-rich foods, perceived as an unpleasant sensation, is often a restricting factor to the consumer´s choices with the daily exclusion of fresh fruits and vegetables from people’s diet. Nonetheless, in Mediterranean diets where polyphenol content can reach up to 1 g/day [14,15,16] polyphenols have become obvious and valuable solutions to be explored in the prevention and management of atherothrombotic diseases and other NCD (diabetes, obesity, and metabolic syndrome). Moreover, from the consumer’s perspective there is an increasing desire for natural products which has sparked research surrounding the health benefits of food polyphenols particularly focused on flavonoids.

Research on the biological effects of food flavonoids has boosted in the past 20 years (Figure 1) and in recent years, in vitro cell studies revealed that the benefits of flavonoids go beyond their antioxidant [17,18,19,20,21] and may also be involved in the regulation of appetite and satiety [22,23,24], inhibition of glucose hydrolytic enzymes and uptake [25,26,27,28], lipid metabolism and lipoprotein function [29,30,31,32], eNOS activity and vascular stiffness [33], and endothelial health [34].

In view of the recognized health-promoting effects of dietary flavonoids as anti-diabetic, anti-hypercholesterolemic, anti-thrombotic, and anti-inflammatory agents [35,36,37] supports the notion that polyphenol-rich diets may serve as nutritional solutions able to exert protective effects in diet-related pathologies (“personalized nutrition”) and as alternative and sustainable strategies to conventional drug-based therapies in the prevention and management of atherothrombotic diseases.

This review describes the current knowledge on the role of polyphenols with emphasis on flavonoids on enzymes and proteins that actively participate in the regulation of lipid metabolism, appetite and lipid accumulation, thrombus formation, and vascular function and as nutritional solution tailored to prevent and manage the risk of atherothrombotic diseases. Finally, we discuss the limitations and challenges associated with current experimental approaches in cell models.

## 2. Role of Mediterranean Diet on Blood Lipids, Lipoproteins, and Their Functionality

Western-type diets rich in processed foods and fats have a deleterious impact on people’s health leading to elevated cholesterol levels, increased obesity and raised blood pressure contributing to the risk of cardio- and vascular complications later in life. However, raised plasma cholesterol is only the “tip of the iceberg” as large-scale lipidomic studies conducted in plasma samples during the past decade [38,39,40,41,42,43,44,45,46,47] have shown that additional changes occur to lipoprotein’s lipid composition. The changes found were related not only to predominant lipid classes such as cholesterol, TAG, and PC, but also to less abundant classes such as sphingolipids (ceramides and sphingomyelins), plasmenyl-PE, sulfatides, and cholesterol sulfate and phospholipids with low unsaturation degree supporting the notion that lipid classes located both in the core and the surface of lipoproteins were involved in lipoprotein’s increased atherogenicity [48].

The in vivo findings have been supported by in vitro studies where the lipid remodeling undergone by oxidatively modified lipoproteins may result in ApoB-100 protein misfolding which is thought to increase the electronegativity at the surface of the particle and lead to aggregation of particles and fusion events to endothelial proteoglycans [49,50,51]. Studies conducted on isolated human LDL bearing atherogenic chemically-(Cu(I), HOCl) and enzymatically-induced modifications (LOX, PLA_2_, SMase) revealed dysfunctional lipid composition when compared to native LDL [49,50,52,53]. Chen and colleagues (1997) reported that the greater proportion of lysoPC lipids, a by-product of PLA_2_ activity, was responsible for the impairment of endothelium-dependent vascular relaxation in hyperlipidemic patients [52]. The oxidative modification of surface lipids, promoted by the inflammatory status of diet-related diseases (hyperglycemia, obesity, and hyperlipidemia), contributes to the formation of oxidized phosphatidylcholines (oxPC) and the increased atherogenicity of lipoproteins. It is now widely accepted that the oxidatively modified acyl chains in oxPL protrude into the aqueous medium (“lipid whisker model” [54]) making these structural motifs physically accessible to actively participate in signaling events. While concentration ranges and spatial distribution of oxPC in lipoproteins in health and disease are not yet known and have been largely overlooked by the scientific community [55], at the biological level oxPC act as danger-associated molecular patterns (DAMP’s) and contribute to the uptake of oxLDL by macrophages into the sub-endothelial space with lipid accumulation in the vessel wall (“foam cells”) an early step in atherogenesis [56]. At the biophysical level, oxPC impact membrane packing and fluidity, lipid lateral organization, permeability, and endothelial stiffness [57,58,59].

To date, although there is a clear understanding of the lipid imbalances to lipoproteins and how these impact lipoprotein atherogenicity and loss of function, there is still limited knowledge on the exact molecular mechanisms by which flavonoids in Mediterranean Diets exert protective effects to lipoproteins and human health. As evidenced by epidemiological human studies, the adoption of polyphenol-rich diets and flavonoid supplementation to our daily diets improves not only obesity markers (BMI, and abdominal circumference) but also cardio- (blood pressure, heart rate, and carotid intima-media thickness) [60,61,62,63,64], vascular (endothelin-1, and NO), and inflammatory markers (hs-CRP, IL-6, and IL-8) in patients with diagnosed NCD, as well as blood lipid composition, namely, plasma cholesterol (total cholesterol and LDL cholesterol) and plasma triglycerides [62,64,65,66,67,68], supporting the notion that food flavonoids have a crucial role in the regulation of lipid metabolism during the liver biosynthesis and intestinal lipid absorption.

In spite of the epidemiological evidences, a more comprehensive understanding of how flavonoids travel in circulation, reach, and cross cells to be able to interact with lipid metabolizing enzymes is vital to understand the benefits of polyphenol-rich diets. Curiously, while the panel of circulating flavonoids in physiological conditions is known [65,69,70,71,72,73,74,75] it is still poorly understood how flavonoids are transported in circulation and their changes in disease. Evidences from the in vitro addition of resveratrol (non-flavonoid) to plasma proteins suggest that polyphenols are preferentially distributed in the lipoprotein fraction (d < 1.21 g/mL) rather than the lipoprotein depleted fraction [76]. This may account for the observed resistance of lipoproteins to metal-catalyzed oxidative damage after flavonoid supplementation. Studies with purified lipoproteins populations have shown that incubation with flavonoids improved the antioxidant capacity of circulating LDL against damaging radical-mediated modifications [77,78,79] thought to occur through a synergism with lipophilic vitamins [80,81,82]. Studies have also shown that flavonoids also interacted with membrane (surface) lipids by increasing polarizability and membrane order during copper-induced oxidation [80] and with ApoB-100 protein which resulted in the increase LDL uptake by macrophages [83]. Currently, there is a strong consensus that flavonoids reside just below the lipid–water interface of lipid membranes [84,85,86,87,88,89,90,91], whereas lipophilic antioxidants (e.g., beta-carotenoids and alpha-tocopherol) immerse deeper into the bilayer providing the synergistic protection to the membrane against radical damage [81]. To date, epidemiological and in vitro studies have pointed out towards a direct association between polyphenol-rich diets and improved health outcomes by reducing the burden of atherogenic lipoproteins. The following sections detail the current knowledge on the effect of polyphenols on risk factors surrounding atherothrombotic diseases.

## 3. Role of Diet Polyphenols on Lipid Metabolism Enzymatic Systems

In humans, low density lipoproteins (LDL) are the main carriers of cholesterol for delivery to peripheral tissues, whereas high density lipoproteins (HDL) transport cholesterol back to the liver. In practice, the ratio of LDL cholesterol (LDL-C) to HDL cholesterol (HDL-C) is a clinical marker widely used to stratify patients at risk of cardiovascular diseases (CVD) events [92]. To cope with the rising incidence of high cholesterol levels (hypercholesterolemia) in western countries and the increased risk of cardio- and vascular complications, lipid-lowering drugs are widely prescribed targeting cholesterol biosynthesis (e.g., statins or fibrates), cholesterol absorption (e.g., ezetimibe) or reverse cholesterol transport (e.g., niacin and CETP inhibitors). Cholesterol homeostasis is vital for proper cellular function and requires a complex cross-talk of many signaling pathways in different cellular organelles designed to maintain normal levels of cholesterol [93]. The expression of HMG-CoA reductase is tightly regulated by cholesterol which under high levels inactivate the expression of sterol regulatory element-binding protein-2 (SREBP-2), a membrane-bound transcription factor, and downregulates HMG-CoA reductase. The membrane transcription factor (SREBP-2) also up-regulates cell-surface LDL receptor (LDL_R_), thus mediating the endocytosis of cholesterol-rich LDL (LDL-C) and hepatic clearance of plasma cholesterol. The levels of LDL_R_ at the surface of cells are regulated by the proprotein convertase subtilisin/kexin type-9 (PCSK9) by their involvement in the lysosomal degradation of the hepatic low density lipoprotein receptors (LDL_R_) [94]. Reduced levels of PCSK9 mean more LDL_R_ are recycled increasing the expression of LDL_R_ on the surface of liver cells, boosting LDL uptake and cholesterol clearance from circulation (Figure 2). Hence, any compounds able to impair the activity of these targets (HMG-CoA, SREBPs, LDL_R_, and PCSK9) are crucial in maintaining cholesterol homeostasis.

At present, although statins are popular lipid-lowering therapy they are counter-indicated for patients with pre-existing liver disease. As an alternative to statins, PCSK9 inhibitors such as evolocumab (Amgen) and alirocumab (Sanofi and Regeneron) have been developed to expand the panel of therapeutical strategies designed to lower cholesterol levels. In fact, it has already been shown that the administration of PCSK9 inhibiting antibody therapy to patients with established coronary heart disease (CHD) significantly altered the lipid composition of plasma [95] without affecting inflammatory markers [96]. The changes observed to lipid composition were mostly attributed to decrease of lipid cargo in LDL and increase lipid content in HDL populations due to marked changes to sphingolipids (glucosylceramides, sphingomyelins, and ceramides) [95] lipid classes that are strongly associated with CVD outcomes [97].

In view of the increasing consumer´s desire for natural products there is a huge effort towards the implementation of nutritional strategies as alternatives to drug-based therapies. Remarkably, several studies have already found that food flavonoids participate and are able to exert regulatory effect in several of the enzymes involved in the metabolism of lipids. In fact, extensive work with both cell and animal models has shown that flavonoids display an inhibitory effect at the cholesterol and fatty acid biosynthesis [29,30,32,98,99,100]. Flavonoids exhibit high binding affinities with HMG-CoA reductase with IC_50_ values within the micromolar range [29,30,99,100]. Based on the IC_50_ values estimated (Table 1) the inhibitory effect to HMG-CoA reductase is related to flavonoid structural features. In silico computational approaches revealed that among 23 plant-derived compounds, rutin (quercetin-glucoside) showed the highest binding affinity between 7 intermolecular H-bonds with key Met534, Cys526, Gly532, and Gln814 that stabilized the flavonoid-protein interaction [101]. Remarkably, while individual flavonoids exhibited hypocholesterolemic properties in cell and animal models [30,98,99,100], studies with flavonoid extracts (mixtures of compounds) obtained from grapefruit, apple, and blackcurrant fruits induced no change to HMG-CoA reductase activity [29,102,103]. Noteworthy, apple polyphenol extract poor in flavonoids (<5%) did not induce significant changes to HMG-CoA, SREBP-2, and LDL_R_ but did suppressed CETP activity [103].

The regulatory effect of flavonoids occurs also at the cholesterol absorption level, where work with flavones (luteolin, apigenin), flavonols (kaempferol, quercetin, and quercetin-glucoside), flavanols (EGCG), isoflavones (genistein), and flavanones (hesperetin, and nobiletin) have all shown to up-regulate the expression of LDL receptor (LDL_R_), a plasma membrane glycoprotein in HepG2 cells [31,107,108,109,110,111]. The increased LDL_R_ expression seems to be related to decrease of gene transcription of SREBP-2 [31,109,110,111] though alternative mechanisms may be involved possibly via ERK signaling pathway [107]. Alternatively, Mbikay and colleagues (2014) proposed that the treatment with quercetin-3-glucoside significantly inhibited sortilin expression at the mRNA and protein levels in Huh7 hepatocytes [31].

The involvement of flavonoids in cholesterol metabolism is also noticeable at transcriptional level by regulating key transcription factors including SREBP-1 [32,112], SREBP-2 [113], and PCSK9 [31,107,114,115]. Working with hepatic WRL and HepG2 cell lines, Wong and colleagues (2015) found that luteolin (flavone) suppressed the expression of SREBP-2 at concentrations as low as 1 μM [113]. This flavone also prevented the nuclear translocation of SREBP-2 by partially blocking post-translational processing through increased AMP kinase (AMPK) activation. At the transcriptional level, the mRNA and protein expression of SREBP-2 were reduced after luteolin treatment [113]. Xanthohumol, a prenylated flavonoid found in hops, suppressed SREBP-1 target gene expression in the liver accompanied by a reduction of the mature form of hepatic SREBP-1 involved in fatty acid synthesis [32]. Similarly, working with low micromolar concentrations of EGCG (flavanol) [107], quercetin-3-glucoside (flavonols) [31], pinostrobin (flavanone) [114] and a flavanone isolated from medicinal Chinese plant *Sculletaria baiacalensis* [115] flavonoids were described to have an inhibitory effect on PCSK9 activity. In spite of the overall consensus reported in the literature, not all compounds displayed the same activity suggesting that flavonoid’s structural features have an impact on inhibitory activity [108,115].

## 4. Dietary Polyphenols as Regulators of Appetite, Tissue Fat Deposition and Obesity

In addition to the anti-cholesterolemic and anti-lipidemic effects mentioned above, diets rich in fruits, vegetables and nuts may also control food intake and energy expenditure by having a direct impact on the regulation of appetite hormones [116,117]. Gut hormones such as cholecystokinin (CCK), glucagon-like peptide (GLP-1), and pancreatic peptide YY_3-36_ (PYY) are satiety hormones released in the blood in response to food regulating appetite and food intake via the “gut”brain axis– [118,119] and their levels are changed upon flavonoid supplementation.

In vitro cell studies conducted in enteroendocrine STC-1 cells with flavanones (naringenin and hesperidin), flavones (apigenin and baicalein) and flavanols (quercetin, kaempferol, and rutin) revealed that these flavonoids stimulated the release of cholecystokinin (CCK) peptide in a concentration-dependent manner [22,23,24]. The same studies also reported that the glycoside forms of flavonoids (naringin, hesperin, and rutin) did not stimulate the release of CCK [22,23,24] though incubation of delphinidin-3-rutoside (anthocyanin) up to 100 μM did stimulate the secretion of GLP-1 in murine GLUTag L cell lines [120].

Remarkably, many of the flavonoids involved in the secretion of appetite hormones were also reported to inhibit pancreatic lipase a key digestive enzyme responsible for the hydrolysis of triglycerides into fatty acids. The inhibition of pancreatic lipase constitutes an anti-obesity strategy as it decreases the absorption of lipids in the small intestine and likely storage of excessive lipids by white adipose tissue [121,122,123]. From the extensive work conducted so far on the inhibitory effect of flavonoids and non-flavonoids on lipase activity, it is apparent that structure plays a pivotal role. Working with apple polyphenol extract, Sugiyama and colleagues reported that catechins, chalcones and benzoic acids displayed weak inhibitory activity whilst procyanidins exhibited the highest inhibitory effect [124]. As oligomeric procyanidins significantly inhibited lipase activity the authors also reported that polymerization was an important factor in the inhibitory activity [124]. These findings are in agreement with others, where among 54 (flavonoid and non-flavonoid) polyphenols from Oolong tea leaves including flavan-3-ols, proanthocyanidins, theaflavins, theasinensis, and hydrolysable tannins, Nakai and colleagues (2005) estimated the IC_50_ values and found that oolongtheanin and theaflavins D and A displayed the strongest inhibitory effect against porcine pancreatic lipase suggesting that galloyl residues were key structural features to the lipase inhibitory effect [122]. Interestingly, in another study although tea catechins (EGCG) displayed higher IC_50_ values when compared to caffeoylquinic dimers (DCQA) found in coffee, DCQA exhibited higher stability during in vitro digestion reaching the proximal intestine which could account for the inhibitory effect of DCQA towards lipase activity [121].

An alternative anti-obesity mechanism by which dietary flavonoids exert health benefits seems to occur at the cell level by regulating the number and size of adipocytes in adipose tissue (adipogenesis). Findings from cell studies show that flavonoids such as EGCG [125] and juglanin [126] and non-flavonoids such as oleuropein and hydroxy-tyrosol [127] inhibited preadipocyte differentiation and reduced adipogenesis in 3T3-L1 cells [125,126,127]. Supplementation with grape seed proanthocyanidins in animal models not only reduced adipocyte size and increased adipocyte number in white adipose tissue (WAT) but also improved levels of serum cholesterol and triglycerides by more than 20% in addition to glucose and insulin levels [128]. Zhu and colleagues (2017) proposed that the strong inhibitory effect of EGCG compounds on preadipocyte differentiation was attributed to the high affinity of catechins (ECG- and EGCG-type) to cholesterol in lipid-rafts which decreased fluidity and the integrity of lipid-raft. The disruption at the cell membrane level suppressed the mitotic clonal expansion process and the expression of mRNA levels of PPARγ, C/EBPα, and SREBP-1c [125]. In another study, Wang and colleagues reported that the reduction of adipogenesis was attributed to inhibition of pro-adipogenic transcription factors (PPARα, PPARγ, C/EBPα, C/EBPβ, and SREBP-1c) through SIRT1/AMPK signaling pathway [126].

While the molecular mechanisms by which flavonoids are able to regulate food intake and tissue fat deposition are far more complex [129], it should also be highlighted that short-chain fatty acids (SCFA), such as propionic acid formed from gut degradation of flavonoids [130] are equally able to stimulate the release of satiety hormones [131,132] leading to reduced weight gain, intra-abdominal adipose tissue distribution, and intrahepatocellular lipid content in overweight adults [132].

Overall, findings from in vitro cell studies have shown that diet flavonoids appear to exert anti-obesity effect by multiple mechanisms namely by modulation of appetite, adipocyte differentiation, size, and number.

## 5. Dietary Polyphenols in the Modulation of Platelet Activation, Cell–Cell Adhesion and Thrombus Formation

The beneficial health effects of flavonoids abundant in Mediterranean-type diets have, up until recently, been mainly related to their free radical scavenger ability and anti-oxidant properties contributing to the maintenance of endogenous cellular redox systems [20,133,134,135,136,137]. However, the involvement of flavonoids on the regulation of lipid metabolizing enzymes (described in Section 3) with subsequent shaping and remodeling of plasma and lipoprotein lipid composition [62,64,65,66,67,68] may be pivotal in the prevention of atherothrombotic events. It is acknowledged that lipoproteins with deregulated lipid composition (oxLDL) trigger the uptake by macrophages and lipid accumulation in the sub-endothelial space leading to foam cell formation [138]. It has recently been reported that lipoprotein’s lipid composition strongly controls aggregation and fusion events reporting that LDL containing more surface sphingolipids and fewer phosphatidylcholines were more prone to aggregation [50]. In another study, variations in core triacylglycerol (TAG) levels influenced the structural stability of VLDL and lipid remodeling increasing the susceptibility of lipoproteins to oxidative modification and fusion events [139]. In view of this, the increased intake of polyphenol-rich foods may reveal to be an additional strategy to control the potential for lipoprotein’s aggregation and fusion events.

The atheroprotective potential of polyphenol-rich diets in the modulation the lipid composition of lipoproteins and their susceptibility to aggregatory events [50,139] may be expanded to contemplate anti-thrombotic effects as several studies have already shown that flavonoid treatment to human blood samples influences platelets activation (thrombocytes), cell–cell aggregation and subsequent clot formation (thrombus). Work with flavanol (epicatechin), flavonol (myricetin), flavone (quercetin) and grape seed extract led to an overall reduction of platelet function parameters as assayed by measurement of platelet aggregation, thrombin formation, clot formation time, and clot firmness [140,141,142,143]. Flavanol treatment also led to decreased adenosine diphosphate (ADP)-induced platelet aggregation, P-selectin expression, decreased platelet thrombin receptor (PAR)-activating peptide-induced aggregation and increased thrombin receptor-activating peptide-induced fibrinogen binding resulting in the inhibition of platelet endothelial cell adhesion molecule-1 (PECAM-1) [140,144,145]. Likewise, work with flavonoid gut metabolite protocatechuic acid (PCA) on isolated human platelets selectively and potently inhibited high shear-stress induced platelet aggregation though it did not inhibit platelet aggregation induced by other endogenous agonists like collagen, thrombin, or ADP also important in pathological thrombosis and normal hemostasis [146].

In addition to the anti-thrombotic and atheroprotective effect of flavonoid intake, the anti-inflammatory effect reported after flavonoid supplementation may also aid and complement the atheroprotective effect. There is an overall consensus that dietary flavonoids are able to suppress the expression of surface pro-inflammatory mediators in cultured endothelial cells such as ICAM-1, VCAM-1, IL-6 and MCP-1 in a structure- and concentration-dependent manner [147,148,149,150,151,152] regardless of the inflammatory stimulus [151,152,153,154,155,156,157,158]. Results obtained with anthocyanins (flavonoids) typically found in berries (grapes, blueberries, strawberries, raspberries) showed that anthocyanins modulated the expression of genes involved in cell–cell adhesion, cytoskeleton organization, or focal adhesion resulting in decreased monocyte adhesion and transendothelial migration [154]. Although the majority of studies has been conducted with flavonoids, the anti-inflammatory activity is also extended to gut flavonoid metabolites [153,156,158,159,160,161]. Lee and colleagues (2017) reported that di-hydroxyphenyl-γ-valerolactone, known flavan-3-ols gut metabolites, prevented the adhesion of THP-1 monocytes to endothelial cells in a dose-dependent manner by down-regulating the expression of VCAM-1 and MCP-1 [156]. In another study, sub-micromolar concentrations of phloroglucinaldehyde (PGA) and protocatechuic acid (PCA) inhibited the production of inflammatory cytokines (IL-6) in cultured LPS-stimulated THP-1 monocytes and macrophages [159].

In overall, the ability of food flavonoids to improve platelet function, inflammatory status, and reduce adherence of blood cells to endothelial cells showcases the health benefits of polyphenol intake to counteract the onset, development and progression of vascular and thrombotic events.

## 6. Dietary Polyphenols in the Regulation of Blood Pressure and VASCULAR Homeostasis

The increased salt intake from processed foods in develop countries has escalated the incidence of individuals with high blood pressure and increased risk of atherothrombotic disorders. While blood pressure is regulated by a complex interplay of systems (i.e., sodium and renin–angiotensin–aldosterone system), angiotensin-converting enzyme (ACE) and nitric oxide (NO), other factors such as arterial stiffness and plaque calcification are equally important factors for assessing vascular function and homeostasis. In fact, coronary artery calcium score (CACS) quantified through imaging techniques is, together with clinical and biochemical parameters (blood pressure, blood glucose, and lipids), used to identify individuals at risk of developing cardiovascular diseases (CVD).

Based on the literature, it is evident that in addition to the anti-hypercholesterolemic, anti-hyperlipidemic, and anti-obesity effects, flavonoid supplementation is able to reduce blood pressure in hypertensive animal models [106,162,163]. The anti-hypertensive effect of plant flavonoids is attributed to inhibition of ACE activity [29,30,104,105,163] and promote the production of vasorelaxating factors (e.g., nitric oxide (NO) release) [162]. In a more exhaustive study conducted on 17 flavonoids tested for the inhibition of ACE activity that included flavanones, flavones, isoflavones, flavonols, and flavanols, all flavonoids inhibited the activity of ACE in a dose-dependent manner with luteolin (flavone) exhibiting the highest inhibitory effect (IC_50_ 23 μM) [104]. In another study with catechins (flavan-3-ols) showed that galloylated catechins exhibited the strongest inhibitory ACE activity (Table 1) than non-galloylated catechins [106]. Complementary structure-activity relationship (SAR) studies determined that the key structural features of tested flavonoids responsible for ACE inhibition was the presence of an unsaturated 2–3 bond conjugated with a 4-oxo- function, aside from the 3′,4′-catechol B-ring pattern [104]. Noteworthy, comparison of ACE inhibitory activity (Table 1) among several catechins (flavanols) and their gut metabolites including valeric acid and valerolactone derivatives revealed that metabolization reduced the inhibition ACE activity and hence the anti-hypertensive potential [106]. These findings are in contrast with a similar study conducted with quercetin (flavonols) and quercetin gut metabolites where the hydroxy-phenylpropionic acid (HPPA) exhibited higher inhibitory ACE activity (Table 1) when compared to quercetin and its methyl derivatives (isorhamnetin and tamarixetin) [164].

As radical scavengers, plant flavonoids also impact the nitrite–nitrate pathway and enhance the production of NO by activation of inducible- and endothelial-nitric oxide synthases (iNOS and eNOS). NO produced from L-arginine in a reaction catalyzed by the nitric oxide synthase (NOS) is a known vasodilator that contributes to endothelial function. Flavonoids have been reported to potentiate vascular function by the activation of endothelial NO synthase [162,165,166] as well as by preserving the integrity of tight junctions and promoting the expression of tight-junction proteins even under inflammatory conditions averting the expression of claudin-2 by inflammatory cytokines [167]. Álvarez-Cilleros found that gut microbial flavanol metabolites increased NO production in human endothelial cells (EA.hy927) after the treatment with 10 µM of 3,4-DHPA or with a mix of that metabolite with 2,3-dihydroxybenzoic acid (2,3-DHB) and 3-hydroxyphenylpropionic acid (3-HPPA) [34]. These findings are in agreement with others reporting that pre-treatment of glucose-challenged human aortic endothelial cells with micromolar amounts (≤5 μM) of quercetin-3-glucuronide (Q3G), piceatannol (PIC), and 3-HPPA preserved NO production even under glycotoxic conditions [168].

Interestingly, and although less studied, microbial flavonoid metabolites such as propionic and butyric acid known as short chain fatty acids (SCFA) [130] are also key modulators of membrane permeability and epithelial barrier function [169,170] and regulators of blood flow [171,172]. The formation of SCFA derived from flavonoids seems to expand the already known beneficial effects of flavonoids, and thus an additional benefit to the implementation of polyphenol-rich diets in the resolution of chronic inflammatory and atherothrombotic disorders.

In summary, aside from coffee where its intake increases the risk of developing hypertension in individuals carrying the 1F allele in CYP1A2 genotype [173], flavonoids included in Mediterranean-type diets (wine, chocolate, tea, and berries) and their metabolites display vasoprotective, vasorelaxant, and anti-hypertensive effects improving vascular homeostasis and health.

## 7. Polyphenol Rich-Diets in the Management of Atherothrombotic Diseases: Current Challenges and Future Directions

Dietary polyphenols, while not essential for life, exert health beneficial effects to humans. Though initially the health benefits were mainly attributed to flavonoids intrinsic antioxidant properties [20,133,136] research conducted in the past decade shows that the benefits associated to polyphenol-rich diets are not just limited to their strong antioxidant capacity contributing to improved plasma redox status [34,174,175,176] but have an array of other benefits. For instance, polyphenols display anti-diabetic properties due to the inhibition of hydrolytic enzymes responsible for glucose degradation and uptake [25,26,28,177], hypocholesterolemic and hypolipidemic properties by the inhibition of lipid metabolism enzymes [29,30,31,32,109], anti-obesity effects by the regulation of satiety hormones and food intake [22,23,24], anti-thrombotic effects by the inhibition of platelet activation and cell–cell aggregation [140,141,142,143]; as well as anti-hypertensive and vasoprotective effects by the regulation of ACE and eNOS [30,162,163,164] contributing to ameliorate the risk of atherothrombotic events and CVD mortality.

Despite the effort to improve our understanding on the relationship between dietary choices and health outcomes, a direct casual effect still remains elusive. The reason behind this may be related to the fact that in most cases researchers have adopted experimental conditions (e.g., tested compounds and concentrations, incubation periods, and cell culture conditions) that are far from those mirroring (patho)physiological conditions, and findings reported so far have little relevance to improve our understanding on the role of polyphenol-rich diets in the prevention and management of atherothrombotic diseases.

### 7.1. Ingested Polyphenols vs. Circulating Polyphenols: Issues of Metabolization and the Occurrence of Metabotypes

Most of our current knowledge on the health benefits of polyphenol-rich diets comes from in vitro experiments conducted with individual polyphenols [24,30,31,100,104,109,111,123,126,127,178,179]. However, though the type and amount of ingested flavonoids is directly related to food processing steps and the consumer’s daily choices, flavonoids are ingested not as individual compounds but as mixtures containing dozens of compounds with different structural features. Further, while the inclusion of flavonoid mixtures (extracts) in cell-based experiments would be more realistic, upon ingestion dietary flavonoids are rapidly metabolized and in vivo they occur as phase II conjugates (e.g., methyl, glucuronide, or sulfate derivatives) and gut metabolites (phase III metabolism) rather than in their precursor form. In this sense, the inclusion of flavonoid metabolites in cell-based studies would be physiologically more relevant rather than the inclusion of flavonoid-rich extracts. Strangely, to date the in vivo metabolization reactions have not been taken into consideration in the many studies investigating the effect of flavonoids on the anti-hypercholesterolemic, anti-hyperlipidemic, anti-hypertensive, anti-obesity, anti-aggregation, and anti-inflammatory properties [99,100,102,109,111,115,121,123,126,127].

Moreover, our current knowledge on the profile of flavonoid metabolites in circulation derives from screening studies on plasma samples collected from healthy normocholesterolemic, normoglycemic and normotensive individuals [65,69,70,71,72,73,74,75] while the panel and cargo of plasma flavonoid metabolites under pathophysiological relevant conditions (e.g., hypercholesterolemia and hyperglycemia) remains elusive. Screening studies focused on deciphering the flavonoid metabolites signature in inflammatory-related conditions (hypercholesterolemia and hyperglycemia) is crucial to better understand the effect of polyphenol-rich diets on endothelial and vascular function. Recent metabolomic studies conducted with urine samples collected in individuals after fruit supplementation evidenced the predominance of specific polyphenol metabolites [71,74,75] and of specific metabolic signatures—metabotypes—in spite of the inherent human genetic inter-individual and microbiota variability. Although findings from these studies were conducted in individuals in “good health” [71,74,75] and were limited by the low sampling included (*n* = 20), the identification of metabotypes in plasma samples in disease conditions has not yet been addressed. In view of this, clustering of individuals depending on their metabotypes to explain flavonoid effects has been overlooked when conducting in vitro cellular studies. It is foreseen that this may certainly impact the concept and design of personalized nutrition guidelines.

### 7.2. Cell Culture Conditions Mimicking Pathophysiological Conditions

The current knowledge on the beneficial health effects of flavonoids derives mainly from in vitro cell-based studies carried out under “healthy” normolipidemic and normoglycemic conditions. Evaluating the cell’s response to flavonoid intake in health has little relevance towards the improved understanding of polyphenol-rich diets in the prevention and management of atherothrombotic diseases.

Only a handful of studies investigated the role of flavonoids in glucose-challenged human umbilical vein endothelial cell (HUVEC) cells [27,150,180,181,182,183]. Though flavonoids induce a protective effect under supra-physiological concentrations of glucose by reducing the expression of ICAM-1, VCAM-1, and E-selectin [150] inhibiting the production of chemotactic MCP-1 protein, the effect of flavonoid metabolites at physiologically relevant conditions of hypercholesterolemia and hyperglycemia remains largely overlooked. Interestingly, pre-treatment of human aortic cells with quercetin-3-O-glucuronide, piceatannol or 3-(3-hydroxy-phenyl)propionoic acid (flavonoid microbial metabolites) prior to high-glucose induced stress prevented elevations in reactive oxygen and nitrogen species in response to high glucose and preserved insulin stimulated increases in NO production [168]. Researchers should make an effort to work under cellular conditions that best mimic pathophysiological conditions and include other oxidative stress-related stimuli likely to occur in inflammatory conditions such as oxPC that impact on endothelial barrier properties [57,184] and oxysterols reported to regulate cholesterol biosynthesis [185].

### 7.3. Polyphenol In Vitro Chemical Stability and In Vivo Residence Time

Most of in vitro experiments found in the literature describing the effect of flavonoids on the modulation of anti-hypercholesterolemic, anti-hyperlipidemic, anti-hypertensive, anti-obesity, anti-aggregation, and anti-inflammatory effects in HepG2 cells use 24 h incubation periods [20,31,107,110,111,168,186,187] which is excessively long bearing in mind the poor chemical stability displayed by flavonoids and non-flavonoids in neutral aqueous solutions [178,188,189,190,191]. Under neutral conditions such as those selected for cell-based studies flavonoids degrade within the first hour of incubation, with some showing complete degradation after 10 min of incubation [190] resulting in significant losses of the parent compound and the co-existence of unmetabolized, metabolized and breakdown products. In view of this, it is questionable whether the measured effect is due to the parent compound or any of the other newly formed (de)conjugated or breakdown species. To mitigate this, it is crucial that cell biologists insert an additional HPLC-MS step in their experimental design to accurately identify and confirm the predominant compound present in the cellular media responsible for the measured effect.

Similarly, cell-based studies conducted with flavonoid metabolites for long incubation periods (>18 h) far exceed the in vivo residence time of flavonoid metabolites in humans and do not reflect physiological conditions. Pharmacokinetic studies have shown that the residence time in circulation hardly exceeds 1–2 h and 6 h for phase II and phase III flavonoid metabolites, respectively, before reaching basal levels [69,192].

### 7.4. Integration of Biological Findings with Biophysical Experiments

Based on the literature available, evidences so far support that, despite their poor antioxidant potential [19,20,149,193], flavonoid metabolites possess anti-inflammatory, anti-diabetic, anti-adhesive, anti-aggregation, and vasoprotective effects [34,152,153,156,160,164,194] holding great potential to mitigate the risk factors of cardio- and vascular complications in western-type diets.

Nevertheless, and in spite of the extensive in vitro research carried out so far, a more comprehensive understanding of how ingested flavonoids travel and reach cells to be able to interact with lipid metabolizing enzymes, reactive oxygen species (ROS) detoxifying systems and vasoprotective enzymatic systems is still elusive. It is widely known that flavonoids interact with lipid bilayers affecting the membrane dynamics and biophysical properties [87,180,195,196]; however, the molecular mechanisms by which flavonoids trigger the biological response remain unclear. The notion that the biological response may be triggered by polyphenol–lipid biophysical interactions is an emerging concept that is gaining increasing popularity. For instance, EGC-type compounds were reported to cluster near cholesterol-rich regions in model membranes which prevented the differentiation of pre-adipocytes [125]. In another study, resveratrol was found to accumulate near sphingomyelin- and cholesterol-enriched membrane regions (lipid rafts) which then triggered activation of downstream intracellular signaling cascade [197]. Moreover, membrane fluidity in HUVECs lost under hyperglycemia conditions was restored after incubation with quercetin, curcumin, and epigallocatechin gallate [180].

To date, the role of flavonoid metabolites on the lipid metabolism (HMG-CoA, SREBP-2, LDL_R_, and others) remains undeciphered, particularly whether flavonoid metabolites directly interact to membrane transcription factor SREBP-2 and membrane protein LDL_R_ or whether flavonoid metabolites interact with membrane lipids and affect the membrane dynamics (organization, fluidity, and packing) affecting indirectly the conformation of membrane proteins and factors. This is particularly crucial considering that apical leaflet of epithelial membranes are cholesterol-rich regions [198] and the affinity of flavonoids decreases with increasing cholesterol content [199]. The inclusion of biophysical data is even more relevant, bearing in mind that advanced lipid end-products (oxPC) formed during inflammatory-related pathologies (hypercholesterolemia and hyperglycemia) are known to affect the membrane’s fluidity, permeability, and endothelial stiffness [57,200,201] and potentially impact the permeation and transport of flavonoid metabolites across the epithelial barrier.

## 8. Conclusions

Fresh fruits and vegetables typical of Mediterranean and Nordic eating habits are rich in bioactive flavonoids and as such have become obvious sources to explore as nutritional strategies to prevent atherothrombotic diseases and manage the risk of CVD.

Findings from in vitro cellular studies are unveiling the multiple health-promoting benefits of polyphenol-rich diets by the modulation of cholesterol biosynthesis, appetite and food intake, with anti-hypertensive and vasoprotecting effects, inhibition of platelet activation, cell–cell adhesion, inflammatory response, and thrombus formation substantiating the improvement in blood lipids and lipoprotein redox status reported in clinical trials and thus effective players in reducing the risk of atherothrombotic events. The reported effects strengthen the use of polyphenol-rich diets as valuable nutritional solutions to prevent atherothrombotic events and help manage the risk of CVD in individuals who have not yet developed symptoms (primary prevention).

However, the associated astringency and bitter taste characteristic of polyphenol-rich foods which is often a restricting factor for the inclusion of fresh fruits and vegetables in people’s diet is challenging to the effective implementation of healthy polyphenol-rich strategies. Added research to the modulation of food acceptance factors (taste, texture, and aroma) may boost the development of functional and tasty polyphenols-enriched foods and the adoption of polyphenol-rich diets.

## Figures and Tables

**Figure 1 jcm-10-01490-f001:**
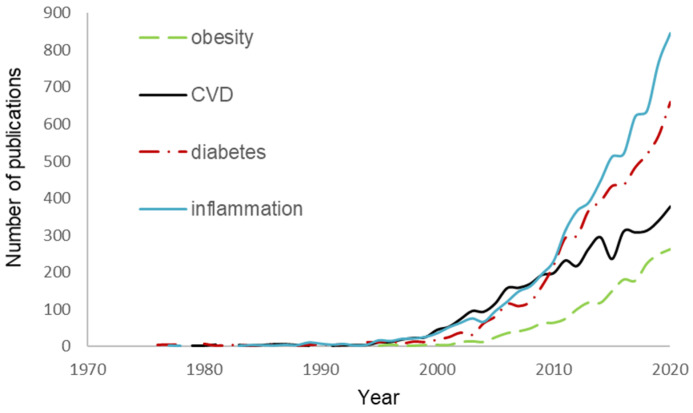
Number of publications using the search string “flavonoids” with other keywords (SCOPUS database, searched 21 December 2020).

**Figure 2 jcm-10-01490-f002:**
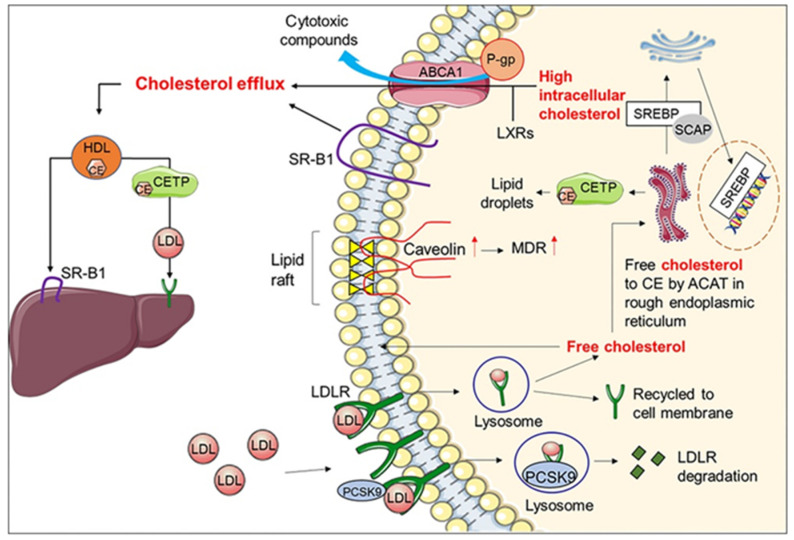
Intracellular and extracellular transport of cholesterol. Extracellular free cholesterol is delivered to various cells via LDL. The LDL then interacts with the LDL_R_ and enters the cells through endocytosis. The free cholesterol is then dissociated from the receptor in the cell lysosome and the unbound receptor is then recycled to the cell membrane for continuous removal of serum cholesterol. However, these LDL_Rs_ undergo lysosomal degradation in the presence of PCSK9. The cholesterol is then distributed within the cells for different functions. Intracellular cholesterol molecules are used to maintain the membrane rigidity through lipid rafts or converted to CEs by ACAT in the endoplasmic reticulum and stored as lipid droplets mediated by CETP. Furthermore, SREBP regulates intracellular cholesterol synthesis and uptake between the endoplasmic reticulum and Golgi body. Intracellular cholesterol homeostasis is maintained through ABCA1, SR-B1, LXR, and caveolins. The coupling of P-gp to ABCA1 regulates cellular toxicity by transporting cytotoxic drugs or compounds to the extracellular matrix. Therefore, ABCA1 is an important protein involved in MDR. In addition, MDR occurs when there is an increase in caveolin (red arrow), which increases membrane cholesterol. Extracellular cholesterol homeostasis is maintained by lipoproteins, HDL and LDLs. Cholesterol is then cleared through HDLs (which interact with SR-B1) to be excreted by the liver or shuttled by CETP from HDLs to the LDLs, which bind to LDL_Rs_ on hepatocytes for cholesterol clearance. (Reprinted with permission from Wiley & Sons from Gu L, Saha TS, Thomas J, Kaur M. “Targeting cellular cholesterol for anticancer therapy” published in FEBS J. (2019) 286, 4192–4208).

**Table 1 jcm-10-01490-t001:** IC_50_ values of polyphenol compounds for 3-hydroxy-3-methylglutaryl-coenzyme A (HMG-CoA) reductase and angiotensin I converting enzyme (ACE) activity.

Enzyme	Polyphenol Compound	IC_50_ Value (μm)	Ref.
***HMG-CoA reductase***	Afzelin	80.1	[30]
	Isoquercitrin	80.6	[30]
	Roxyloside A	47.1	[30]
	Quercetin gentiobioside	50.6	[30]
***ACE***	Apigenin K	196	[104]
	Apigenin-7-O-glucoside	183	[104]
	Luteolin	23	[104]
	Quercetin	43	[104]
	Quercetin-3-O-glucoside	64	[104]
	Kaempferol	178	[104]
	Gallocatechin	195.9	[105]
	Catechin-3-O-gallate	113.0	[105]
	Epicatechin-3-O-gallate	18.3	[105]
	Epigallocatechin-3-O-gallate	37.4	[105]
	Epigallocatechin-3-O-methylgallate	26.6	[105]
	Cyanidin-3-O-glucoside	138.8	[30]
	1,2,3,6-tetra-galloyl-glucose	101.4	[105]
	1,2,3,4,6-penta-galloyl-glucose	73.1	[105]
	5-(3,4,5-trihydroxyphenyl)-γ-valerolactone	2890	[106]
	5-(3,5-dihydroxyphenyl)-γ-valerolactone	19590	[106]
	4-hydroxy-5-(3,4,5-trihydroxyphenyl)-valeric acid	5410	[106]
	4-hydroxy-5-(3,5-dihydroxyphenyl)-valeric acid	12120	[106]
	5-(3,4,5-trihydroxyphenyl)-valeric acid	1510	[106]
	5-(3,5-dihydroxyphenyl)-valeric acid	2380	[106]
	5-(3-hydroxyphenyl)-valeric acid	3000	[106]

## Data Availability

The data presented in this study are available in the cited references.

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
