# Peer review of "Going “Green” in the Prevention and Management of Atherothrombotic Diseases: The Role of Dietary Polyphenols"

_jcm, 2021, doi:10.3390/jcm10071490_

Round 1
Reviewer 1 Report
The manuscript is relatively interesting due to taking into account the general effects of polyphenols and in particular flavonoids included in the Mediterranean type Diet on many regulation aspects of cardio and vascular disease. However, the authors should focus more on flavonoids action (as it seemed to be the main goal of the manuscript) and less on the general effects of polyphenols. Or at least assume this point for polyphenols and do not put the target on the flavonoids in particular. Throughout the text I would expect to find information on flavonoids and ended up finding only information on polyphenols, in an unclear mix
Moreover, one of the biggest problems of the manuscript is due to the abbreviations. Sometimes the abbreviations are defined, sometimes are not, or even sometimes are defined twice. I suggest a careful revision to harmonize and improve the text readability and if possible include a list of abbreviations.
Another problem I found throughout the text was formatting problems, especially regarding special characters. It was sometimes impossible to identify and understand the units or Greek letters associated with the terms.
Detailed comments:
Section 3 is the one that seems to be the least successful overall. Too much information, but not enough detail and badly organised.
Initially, the authors talk about appetite regulation, with some hormones and this topic is brought up again at the end of this section with leptin, with other unrelated targets in between. Needs revision. Moreover, as the section claims the effects of flavonoids on the targets and since this section is named "flavonoids" I would expect to find more direct information on flavonoids and not on polyphenols in general. This topic was extensively revised at Rufino AT, Costa VM, Carvalho F, Fernandes E. Flavonoids as antiobesity agents: A review. Med Res Rev. 2021 Jan;41(1):556-585 and at Song D, Cheng L, Zhang X, Wu Z, Zheng X. The modulatory effect and the mechanism of flavonoids on obesity. J Food Biochem. 2019 Aug;43(8):e12954.
On the section 4 for the authors claims that "However, the involvement of flavonoids on the regulation of lipid metabolizing enzymes (described in section 3)" I do not found the topic of the enzymes of lipid metabolism described up there. Please revise.
section 4: lines 328/329, the authors claim " suggesting that flavonoids may control the potential for lipoprotein’s aggregation and fusion events." But I cannot find this evidence on the reference cited.
Section 5: First paragraph lacks of references.
I think that the manuscript would benefit from an extensive English revision as sometimes the sentences are too long and difficult to read. (e.g. section 6 fist paragraph) and there are many spelling or grammatical errors.
Section 6.1: lines 460-465. Be careful with the assertivity of the sentences. I understand the statement, but I think the use/study of isolated compounds has more to do with their pharmacological potential (which may be lost with the extract) than with their ability to exert beneficial effects after ingestion through the diet in the quantities necessary.
Author Response
We would like to thank the reviewer for his/her valuable and constructive comments and suggestions made. We have addressed all issues raised by the reviewer, which we feel have greatly strengthened the manuscript.
Below you’ll find our reply to the reviewer’s comments and concerns. The additional information requested by the reviewer is described below and was included throughout the manuscript, whilst also aiming not to increase the length of the paper too much. The changes made along the text were highlighted throughout the text for easier tracking.
The manuscript is relatively interesting due to taking into account the general effects of polyphenols and in particular flavonoids included in the Mediterranean type Diet on many regulation aspects of cardio and vascular disease. However, the authors should focus more on flavonoids action (as it seemed to be the main goal of the manuscript) and less on the general effects of polyphenols. Or at least assume this point for polyphenols and do not put the target on the flavonoids in particular. Throughout the text I would expect to find information on flavonoids and ended up finding only information on polyphenols, in an unclear mix.
Reply: Agree. Although initially we were aiming to emphasize the action of flavonoids on cardio and vascular diseases, the inclusion of findings obtained with non-flavonoid compounds may lead to confusions particularly among the non-expert audience. For this reason, we have rephrased the title to “Going “green” in the prevention and management of atherothrombotic diseases: the role of dietary polyphenols”. For the sake of clarity we have also rephrased the title of sections.
Moreover, one of the biggest problems of the manuscript is due to the abbreviations. Sometimes the abbreviations are defined, sometimes are not, or even sometimes are defined twice. I suggest a careful revision to harmonize and improve the text readability and if possible include a list of abbreviations.
Reply: Agree. Following the reviewer suggestion, we have thoroughly revised the manuscript and included an abbreviation list at the end of the manuscript (page 11).
Another problem I found throughout the text was formatting problems, especially regarding special characters. It was sometimes impossible to identify and understand the units or Greek letters associated with the terms.
Reply: Agree. We have corrected the formatting issues found throughout the manuscript.
Detailed comments:
Section 3 is the one that seems to be the least successful overall. Too much information, but not enough detail and badly organised.
Initially, the authors talk about appetite regulation, with some hormones and this topic is brought up again at the end of this section with leptin, with other unrelated targets in between. Needs revision. Moreover, as the section claims the effects of flavonoids on the targets and since this section is named "flavonoids" I would expect to find more direct information on flavonoids and not on polyphenols in general. This topic was extensively revised at Rufino AT, Costa VM, Carvalho F, Fernandes E. Flavonoids as antiobesity agents: A review. Med Res Rev. 2021 Jan;41(1):556-585 and at Song D, Cheng L, Zhang X, Wu Z, Zheng X. The modulatory effect and the mechanism of flavonoids on obesity. J Food Biochem. 2019 Aug;43(8):e12954.
Reply: We agree that the topic is in itself very interesting and deserves a more comprehensive description of the molecular mechanisms. However, within the topic of the Special Issue on Lipids, Lipoproteins and Atherothrombotic diseases we were mainly aiming at highlighting the biological aspects by which polyphenol-rich diets can exert anti-obesity effects particularly in the regulation of appetite and fat tissue deposition. Following the reviewer’s suggestions we have removed the sentence referring to leptin which is out of place as well as the associated reference from the reference list (Ibars et al., 2017, Int J Obes 41, 129-136). Also, we have included the reference by Rufino et al., Med Res Rev. 2021 Jan;41(1):556-585 in the reference list for those who may be particularly interested on the subject of flavonoids as modulators of obesity.
On the section 4 for the authors claims that "However, the involvement of flavonoids on the regulation of lipid metabolizing enzymes (described in section 3)" I do not found the topic of the enzymes of lipid metabolism described up there. Please revise.
Reply: Section 3 gives an overall view on the lipid metabolizing pathways at the level of cholesterol biosynthesis (e.g. HMG-CoA), cholesterol absorption (e.g. LDLR and PCSK9) or reverse cholesterol transport (e.g. CETP) and briefly describes the findings on the effect of flavonoids on these enzymes as well as in lipase activity. Based on the literature available flavonoids are known to inhibit HMG-CoA activity, decrease the activity of SREBP-2, expression of surface LDLR and PCSK9 activity conferring polyphenol-rich diets with an hypocholesterolemic and hypolipidemic effect.
section 4: lines 328/329, the authors claim " suggesting that flavonoids may control the potential for lipoprotein’s aggregation and fusion events." But I cannot find this evidence on the reference cited.
Reply: Agree. We have restructured the sentence as following the statement “…variations in core triacylglycerol (TAG) levels influenced the structural stability of VLDL and lipid remodeling increasing the susceptibility of lipoproteins to oxidative modification and fusion events [136]…” the sentence, as is, induces the reader on the idea that based on the work by Jayaraman et al., (2019) flavonoids may control the potential for lipoprotein’s aggregation and fusion. This is not the case. We believe that by modulating the composition of blood lipids polyphenol-rich diets may exert beneficial effects by regulating lipoprotein’s aggregation and fusion.
Section 5: First paragraph lacks of references.
Reply: Agree. References were included.
I think that the manuscript would benefit from an extensive English revision as sometimes the sentences are too long and difficult to read. (e.g. section 6 fist paragraph) and there are many spelling or grammatical errors.
Reply: We have revised the manuscript with the assistance of a UK native speaker.
Section 6.1: lines 460-465. Be careful with the assertivity of the sentences. I understand the statement, but I think the use/study of isolated compounds has more to do with their pharmacological potential (which may be lost with the extract) than with their ability to exert beneficial effects after ingestion through the diet in the quantities necessary.
Reply: In fact, many of the known beneficial effects of dietary polyphenols derive from in vitro experiments (first sentence of this section). As the main focus of this manuscript was to assess the potential of polyphenol-rich diets in the prevention, management and treatment of atherothrombotic diseases, and not on the pharmacological effects described for the individual polyphenols in the various cell models, we have tentatively included findings reported in the literature on individual, mixtures (extracts) as well as polyphenol metabolites. While searching the literature we found that the majority of studies found in the literature use parent compounds found in plant-based foods. As upon ingestion, polyphenols are rapidly metabolized we believe that many of the findings reported in the literature are not easily translatable to physiological conditions and find little relevance in atherothrombotic scenarios prompting the need for additional cell-based studies on polyphenol metabolites (as highlighted in lines 460-465 from this section).
Reviewer 2 Report
Please, you can find the reviewer comments and suggestions in the attached word document.

Author Response
We would like to thank the reviewer for his/her valuable and constructive comments and suggestions made. We have addressed all issues raised by the reviewer, which we feel have greatly strengthened the manuscript.
Below you’ll find our reply to the reviewer’s comments and concerns. The additional information requested by the reviewer is described below and was included throughout the manuscript, whilst also aiming not to increase the length of the paper too much. The changes made along the text were highlighted throughout the text for easier tracking.
Dear Editor, dear Authors, Ana Reis et al. submitted a review paper based on the role of flavonoids in the prevention of atherothrombic diseases. The review is well-written covering wide range of literature related to the field. Thus, I believe that the manuscript can be accepted for publication in Journal of Clinical Medicine. However, I have some comments to the authors that should be revised before acceptance.
Comments to Author:
First of all, you should check the importance of each section and the format of the journal. For example, the titles: “Role of Mediterranean diet on blood lipids, lipoproteins and their functionality” (line 89) and “Role of diet flavonoids on lipid metabolism enzymatic systems” (line 154) have the same importance as title 1 (line 26)?
Check the number of the title that appear in lines 243-244? Do you refer to section 3 or 4? The same occurs with the lines 313-314, lines 371-372, lines 431-432, lines 454-455, line 489, line 509, line 563. Do the conclusions correspond to section nº 8 (and not to section nº 5)?
Reply: Agree. Sections were misnumbered and this is now corrected.
Moreover, I have some minor corrections that I expose below:
- I suggest to add the full name after some common abbreviations when you mentioned them for the first time in the text:
Line 11: CVD (Cardiovascular diseases)
Line 37: EU (European Union)
Line 38: GDP (Gross Domestic Product)
Line 76: NCD (Noncommunicable diseases)
Reply: As abbreviations such as CVD, EU, GDP and NCD are widely accepted and used in the literature, these were not included in the abbreviation list, unlike the more specific and technical abbreviations (point below). However, for the sake of clarity, we have included them and others in the abbreviation list (page 11).
- What are eNOS (line 84), TAG (line 95), PC (line 96), plasmenyl-PE (line 97), LOX, PLA (line 105), oxPL (line 112), BMI (line 125), NO (line 126), hs-CRP, IL-6, IL-8 (line 127), CEs, ACAT (line 182), CETP (line 183), ABCA1, SR-B1, LXR, P-gp (line 185), MDR (line 186), EGCG (line 218), ECG (line 291), PPAR-α, C/EBP-α (line 294), PPAR-γ, C/EBP-β (line 296), SIRT1/AMPK signaling pathway (line 297), PYY, GLP-1 (line 305), VLDL (line 327), ICAM-1, VCAM-1, MCP-1 (line 352), ACE activity (line 385), iNOS (line 405), 3,4-DHPA (line 412), HUVEC (line 496), ROS (line 537)? You mention directly the abbreviations without the full term (I recommend you to add a list of the abbreviations with their full names. You can add it in the supporting information). Take as an example the list of abbreviations with their full names of the article that you have referenced in your work: “Targeting cellular cholesterol for anticancer therapy”).
Reply: Agree. The abbreviation list was missing from the original submission and is now included in the revised manuscript (page 11).
- The quality of Figure 1 is not very good. Please prepare it with high resolution and add a different color to each curve to make it more attractive.
Reply: Agree. Following the reviewer’s suggestion, the figure was changed and included in a high-resolution format (.png format).
- Line 92-94: However, raised plasma cholesterol is only the “tip of the iceberg” as large-scale lipidomic studies conducted in plasma samples during (in) the past decade [38-47] have shown that additional changes occur to lipoprotein’s lipid composition.
I consider to replace the second “in” by “during”.
Reply: Agree.
- Line 133-135: Curiously, while the panel of circulating flavonoids in physiological conditions is known [65,69-75], it still remains poorly understood how flavonoids are transported in circulation and their changes in disease. (remains poorly understood). The sentence is clearer and more understandable than in the text.
Reply: Agree.
- Line 213: You mention IC50 (Half maximal inhibitory concentration). The correct expression is IC50 (The number 50 should appear as subindex). Please change it always in the manuscript, please.
Reply: Agree. The notation was corrected throughout the text.
- Line 255-259: The same studies also reported that the glycoside forms of flavonoids (naringin, hesperin and rutin) did not stimulate the release of CCK [22-24] though incubation of delphinidin-3-rutoside (anthocyanin) up to 100M did stimulate the secretion of GLP-1 in murine GLUTag L cell lines [117]. Do you refer to 100 mM or 100 µM?
Reply: This is a formatting error and is now corrected.
- Lines 276-280: Interestingly, in another study although tea catechins (EGCG) displayed higher IC50 values when compared to caffeoylquinic dimers (DCQA) found in coffee, DCQA exhibited higher stability during in vitro digestion reaching the proximal intestine which could account for the inhibitory effect of DCQA towards lipase activity [118].
What letters do the initials DCQA correspond to? Maybe you should mention caffeoylquinic acid dimers?
Reply: This is the author’s original notation (as mentioned in the original manuscript Cha et al., J Agric Food Chem 2012, 60, 7152-7157) hence why we have not changed it. For the sake of clarity, the DCQA abbreviation was included to the abbreviation list.
- Lines 292-294: The disruption at the cell membrane level suppressed the mitotic clonal expansion process and the expression of mRNA levels of PPAR, C/EBP and SREBP-1c [122]. Check if you refer to PPAR-α or PPAR-γ; C/EBP-α or C/EBP-β.
Reply: The sentence refers to PPAR-g, C/EBP-a. This is now corrected.
- Lines 386-390: In a more exhaustive study conducted on 17 flavonoids tested for the inhibition of ACE activity that included flavanones, flavones, isoflavones, flavonols and flavanols, all flavonoids inhibited the activity of ACE in a dose-dependent manner with luteolin (flavone) exhibiting the highest inhibitory effect (IC50 23 M) [162]. Do you refer to 23 mM or 23 µM?
Reply: Agree. This is now corrected.
- Lines 414-417: These findings are in agreement with others reporting that pretreatment of glucose-challenged human aortic endothelial cells with micromolar amounts (5M) of quercetin-3-glucuronide (Q3G), piceatannol (PIC) and 3-HPPA preserved NO production even under glycotoxic conditions [168]. Do you refer to < 5 µM?
Reply: Agree. This is now corrected.
- Lines 434-446: This sentence is very long in difficult to follow. You should split it.
Reply: Agree. The sentence was restructured.
- Lines 482-485: Although findings from these studies were conducted in individuals in “good health” [71,74,75] and were limited by the low sampling included (n 20), the identification of metabotypes in plasma samples in disease conditions has not yet been addressed. Do you refer to n = 20? Or <20? Please add the correct symbol.
Reply: Agree. This is now corrected.
- Lines 512, 525 and 527: 24hr, 18hr, 1-2 hours and 6 hours. Please, use the same format for the term “hours”.
Reply: Agree. This was corrected along the text.
Finally, you should correct some mistakes found in the following References:
Reply: All references (below) were corrected.
- E, W.; L, W.; K, W.; P, B.; J, L.; R, L.-F.; R, B.; M, R.; N, T. European Cardiovascular Diseases Statistics 2017 European Heart Network, Brussels 2017.
You should correct the surname of the authors and add here.
- Force, U.S.P.S.T.; Bibbins-Domingo, K.; Grossman, D.C.; Curry, S.J.; Davidson, K.W.; Epling, J.W., Jr.; Garcia, F.A.R.; Gillman, M.W.; Kemper, A.R.; Krist, A.H., et al. Statin Use for the Primary Prevention of Cardiovascular Disease in Adults: US Preventive Services Task Force Recommendation Statement. JAMA 2016, 316, 1997-2007.
You should eliminate the first name because it does not correspond to an author.
- Anuar, M.N.N. Determination of total phenolic, flavonoid content and free radical scavenging activities of common herbs and spices. J Pharmacogn Phytochem 2014, 3, 104-108.
You should add the page numbers and the Abbreviation of the Journal of Pharmacognosy and Phytochemistry.
- Mundra, P.A.; Barlow, C.K.; Nestel, P.J.; Barnes, E.H.; Kirby, A.; Thompson, P.; Sullivan, D.R.; Alshehry, Z.H.; Mellett, N.A.; Huynh, K., et al. Large-scale plasma lipidomic profiling identifies lipids that predict cardiovascular events in secondary prevention. JCI Insight 2018, 3, e121326.
You should add the article number like in some other references.
- Öörni, K.; Hakala, J.K.; Annila, A.; Ala-Korpela, M.; Kovanen, P.T. Sphingomyelinase Induces Aggregation and Fusion, but Phospholipase A<sub>2</sub> Only Aggregation, of Low Density Lipoprotein (LDL) Particles: TWO DISTINCT MECHANISMS LEADING TO INCREASED BINDING STRENGTH OF LDL TO HUMAN AORTIC PROTEOGLYCANS *. J Biol 707 Chem 1998, 273, 29127-29134.
Revise the title of this article.
- Reis, A. Oxidative Phospholipidomics in health and disease: Achievements, challenges and hopes. Free Radic Biol Med 2017, 111, 25-37.
You should add the page numbers.
- Odai, T.; Terauchi, M.; Kato, K.; Hirose, A.; Miyasaka, N. Effects of Grape Seed Proanthocyanidin Extract on Vascular Endothelial Function in Participants with Prehypertension: A Randomized, Double-Blind, Placebo-Controlled Study. Nutrients 2019, 11, 2844-2855.
You should add the article number like in some other references.
- Vitale, M.; Masulli, M.; Calabrese, I.; Rivellese, A.A.; Bonora, E.; Signorini, S.; Perriello, G.; Squatrito, S.; Buzzetti, R.; Sartore, G., et al. Impact of a Mediterranean Dietary Pattern and Its Components on Cardiovascular Risk Factors, Glucose Control, and Body Weight in People with Type 2 Diabetes: A Real-Life Study. Nutrients 2018, 10, 1067.
You should add the article number like in some other references.
- Belguendouz, L.; Frémont, L.; Gozzelino, M.T. Interaction of transresveratrol with plasma lipoproteins. Biochemical Pharmacology 1998, 55, 811-816.
Revise the journal abbreviation: Biochem. Pharmacol.
- Atrahimovich, D.; Khatib, S.; Sela, S.; Vaya, J.; Samson, A.O. Punicalagin Induces Serum Low-Density Lipoprotein Influx to Macrophages. Oxid Med Cell Longev 2016, 2016, 7124251.
- Kitamura, K.; Okada, Y.; Okada, K.; Kawaguchi, Y.; Nagaoka, S. Epigallocatechin gallate induces an up-regulation of LDL receptor accompanied by a reduction of PCSK9 via the annexin A2-independent pathway in HepG2 cells. Mol Nutr Food Res 2017, 61, 1600836.
You should add the article number like in some other references.
- Stote, K.; Corkum, A.; Sweeney, M.; Shakerley, N.; Kean, T.; Gottschall-Pass, K. Postprandial Effects of Blueberry (Vaccinium angustifolium) Consumption on Glucose Metabolism, Gastrointestinal Hormone Response, and Perceived Appetite in Healthy Adults: A Randomized, Placebo-Controlled Crossover Trial. Nutrients 2019, 11, 202.
You should add the article number like in some other references.
- Cruciani, S.; Santaniello, S.; Garroni, G.; Fadda, A.; Balzano, F.; Bellu, E.; Sarais, G.; Fais, G.; Mulas, M.; Maioli, M. Myrtus Polyphenols, from Antioxidants to Anti-Inflammatory Molecules: Exploring a Network Involving Cytochromes P450 and Vitamin D. Molecules 2019, 24, 1515.
You should add the article number like in some other references.
- Bijak, M.; Sut, A.; Kosiorek, A.; Saluk-Bijak, J.; Golanski, J. Dual Anticoagulant/Antiplatelet Activity of Polyphenolic Grape Seeds Extract. Nutrients 2019, 11, 93.
You should add the article number like in some other references.
- Lee, C.C.; Kim, J.H.; Kim, J.S.; Oh, Y.S.; Han, S.M.; Park, J.H.Y.; Lee, K.W.; Lee, C.Y. 5-(3',4'-Dihydroxyphenyl-gamma-valerolactone), a Major Microbial Metabolite of Proanthocyanidin, Attenuates THP-1 Monocyte-Endothelial Adhesion. Int J Mol Sci 2017, 18, 1363.
You should add the article number like in some other references.
- Liu, Z.; Nakashima, S.; Nakamura, T.; Munemasa, S.; Murata, Y.; Nakamura, Y. (-)-Epigallocatechin-3-gallate inhibits human angiotensin-converting enzyme activity through an autoxidation-dependent mechanism. J Biochem Mol Toxicol 2017, 31, e21932.
You should add the article number like in some other references.
- Huang, W.; Hutabarat, R.P.; Chai, Z.; Zheng, T.; Zhang, W.; Li, D. Antioxidant Blueberry Anthocyanins Induce Vasodilation via PI3K/Akt Signaling Pathway in High-Glucose-Induced Human Umbilical Vein Endothelial Cells. Int J Mol Sci 2020, 21, 1575.
You should add the article number like in some other references.
- Piwowar, A.; Rorbach-Dolata, A.; Fecka, I. The Antiglycoxidative Ability of Selected Phenolic Compounds-An In Vitro Study. Molecules 2019, 24, 2689.
You should add the article number like in some other references.
- Verzelloni, E.; Pellacani, C.; Tagliazucchi, D.; Tagliaferri, S.; Calani, L.; Costa, L.G.; Brighenti, F.; Borges, G.; Crozier, A.; Conte, A., et al. Antiglycative and neuroprotective activity of colon-derived polyphenol catabolites. Mol Nutr Food Res 2011, 55 Suppl 1, S35-43.
You should suppress: “Suppl 1”.
Reviewer 3 Report
This is an extensive review of the many medicinal properties of flavonoids. Most important, this review dives into the molecular mechanism of actions and the limitations of in vitro and in vivo studies involving flavonoids. The only other pharmacokinetic property of flavonoids that I would have loved to see mentioned in this article is their solubility and its contribution to the oral bioavailability of naturally occurring flavonoids. Maybe that can be considered for a separate review article.
Author Response
We would like to thank the reviewer for his/her valuable and constructive comments and suggestions made. We have addressed all issues raised by the reviewer, which we feel have greatly strengthened the manuscript.
Below you’ll find our reply to the reviewer’s comments and concerns. The additional information requested by the reviewer is described below and was included throughout the manuscript, whilst also aiming not to increase the length of the paper too much. The changes made along the text were highlighted throughout the text for easier tracking.
This is an extensive review of the many medicinal properties of flavonoids. Most important, this review dives into the molecular mechanism of actions and the limitations of in vitro and in vivo studies involving flavonoids. The only other pharmacokinetic property of flavonoids that I would have loved to see mentioned in this article is their solubility and its contribution to the oral bioavailability of naturally occurring flavonoids. Maybe that can be considered for a separate review article.
Reply: Agree. The bioavailability of flavonoids and other polyphenols is largely dependent on their solubility in aqueous media (e.g. blood plasma) which relates to their chemical features, as well as their permeability in lipophilic environments (e.g. cell membranes). While it is known that the delivery (bioavailability) of ingested polyphenols to surrounding tissues occurs through passive diffusion and active transport, both of which are influenced by the membrane’s lipid composition. This is an exciting subject that we are currently investigating in our group and rather challenging considering that apical and basolateral leaflets of endothelial cells contain distinct content of cholesterol (reference 198 of this manuscript).
Round 2
Reviewer 1 Report
I found the author's response and revised version acceptable publication.